# Histopathological profile of cervical punch biopsies and risk factors associated with high-grade cervical precancerous lesions and cancer in northwest Ethiopia

Awoke Derbie[1,2,3,4]*, Bereket Amare[5], Eyaya Misgan[6], Endalkachew Nibret[3,7], Melanie Maier[8], Yimtubezinash Woldeamanuel[2,4], Tamrat Abebe[4]

1 Department of Medical Microbiology, College of Medicine and Health Sciences, Bahir Dar University, Bahir Dar, Ethiopia, 2 Centre for Innovative Drug Development and Therapeutic Trials for Africa (CDT-Africa), College of Health Sciences, Addis Ababa University, Addis Ababa, Ethiopia, 3 Department of Health Biotechnology, Institute of Biotechnology, Bahir Dar University, Bahir Dar, Ethiopia, 4 Department of Medical Microbiology, Immunology and Parasitology, School of Medicine, College of Health Sciences, Addis Ababa University, Addis Ababa, Ethiopia, 5 Department of Pathology, College of Medicine and Health Sciences, Bahir Dar University, Bahir Dar, Ethiopia, 6 Department of Gynecology and Obstetrics, College of Medicine and Health Sciences, Bahir Dar University, Bahir Dar, Ethiopia, 7 Department of Biology, College of Science, Bahir Dar University, Bahir Dar, Ethiopia, 8 Department of Diagnostics, Institute of Virology, Leipzig University Hospital, Leipzig, Germany

* awoke.derbie@bdu.edu.et

**Data Availability Statement:** All relevant data are within the paper.

## Abstract

### Introduction

Cervical cancer is an important public health problem in Ethiopia. However, the disease is not well characterized and studied in various parts of the country. This study was designed to describe the histopathological profile of cervical biopsies and to identify risk factors associated with high-grade cervical lesions and cancer (CIN2+C).

### Methods

A cross-sectional study was conducted at Felege Hiwot Compressive Specialized Hospital (FHCSH) between 1 March 2019 and 30 October 2021. A structured questionnaire was used to collect data on the participants' demographic, reproductive and gynecologic history. From women presented with different degrees of cervical lesions, a senior gynecologist collected cervical swabs using (Digene HC2 DNA collection device: Qiagen, Hilden, Germany) for detection of high-risk Human papillomaviruses (HR-HPV) and punch biopsy for histopathological examinations. HR-HPVs were detected using the Abbott Alinity m system following the manufacturer protocol at the Institute of Virology, Leipzig University Hospital, Germany. Collected data entered and analyzed using SPSS version 25. A logistic regression model was used for both bivariable & multivariable analysis in order to determine the association between independent variables and CIN2+C. Statistical significance was set at a *p*-value <0.05.

**Funding:** The study was financially supported by Centre for Innovative Drug Development and Therapeutic Trials for Africa (CDT-Africa), College of Health Sciences, Addis Ababa University, Ethiopia and Biotechnology Research Institute, Bahir Dar University Ethiopia. The funders had no role in study design, data collection and analysis, decision to publish, or preparation of the manuscript.

**Competing interests:** The authors have declared that no competing interests exist.

## Results

In this study, 335 women were included; the mean age was at 46.5±11.4 years. Most were living in rural settings, 221(66%) and had no formal education, 259 (77.3%). More than half of the participants, 193(57.6%) were unaware of cervical cancer. The prevalence of HIV infection and previous history of cervical screening were 44(13.1%) and 93(27.8%), respectively. HR-HPVs were detected in 178(54.3%) of the participants. The majority of biopsies, 140(41.8%; 95%CI: 36.6–47.1%), were diagnosed as cervical carcinoma. Normal histology, cervicitis, cervical intraepithelial neoplasia (CIN)-1, CIN-2, and CIN-3 accounted for 74 (22.1%), 30(9.0%), 40(11.9%), 12(3.6%), and 12(3.6%), respectively. High-grade lesions and cancer (CIN2+C) together accounted 164(49.0%; 95%CI: 43.6–54.2). Cervical cancer increased steadily with the age of the participants ($p$<0.001) in which women above the age of fifty were approximately four times more likely to develop CIN2+C than the younger ones (AOR: 3.73; 95%CI: 1.80–7.82; $p$<0.001). Likewise, no screening history in the last five years (AOR: 2.03; 95%CI: 1.05–3.92; $p = 0.035$) and being infected with HR-HPVs (AOR: 14.23; 95%CI: 7.9–25.64; p<0.001) were found significantly associated with CIN2+C.

## Conclusions

The findings of this study revealed that cervical cancer continues to be an important women's health challenge in northwest Ethiopia. Postmenopausal women, who had no screening history within a five-year period and those women who tested positive for HR-HPV need special attention. It is important to increase the awareness of women about cervical cancer and actions for early detection of precancerous lesions should be expanded.

## Background

Disease in the uterine cervix may range from cervicitis to invasive cancerous lesions. Cervical cancer is one of the most common cancers among women globally representing about 8% of all female cancer deaths every year [1]. According to the 2020 world HPV information center report, approximately 604,127 women were diagnosed with cervical cancer and 341,831 die from the disease annually [1,2]. In 2030, the annual number of new cases and death are projected to upsurge to 700, 000 and 400, 000, respectively [3,4]. Close to 90% of the newly diagnosed cases and deaths were from low and middle-income countries [1]. The highest burden of the disease is reported in sub-Saharan Africa [3,5–7].

Cervical cancer is one of the emerging public health challenges in Ethiopia. According to the international agency for research on cancer assessments, the estimated number of new cervical cancer cases at 7,500 in 2020 could intensify to 15,300 in 2040 in Ethiopia. Similarly, the mortality from the disease could increase from 5,340 in 2020 to 11,000 in 2040 yearly in Ethiopia [4]. The country has started vaccinating school girls recently and visual inspection of the cervics with acetic acid (VIA) is the main method of cervical screening for women above the age of 30.

Persistent infection with oncogenic or high-risk Human papillomaviruses (HR-HPV; e.g., HPV16, 18, 31, 33, 35, 39, 45, 51, 52, 56, 58, 59 and 68) is the main cause of cervical cancer [8]. However, a recent study revealed that nearly 5% of cervical cancer cases were not HPV-associated, which is a unique biological entity with a different molecular profile when compared with HPV-associated cancers [2].

Cervical intraepithelial neoplasia (CIN) is a premalignant lesion that may exist in either of these stages: low-grade (CIN1) or high-grade (CIN2 or CIN3) depending on the degree of epithelial tissue formed by proliferative lesions [9]. CIN describes how far the abnormal cells have gone into the surface layer of the cervix. It is classified into grades based on the cell changes. These grades reflect how much of the cervix is affected by the abnormal cells. In CIN1, proliferative lesions could form the lower third of the epithelial tissue. In CIN2, proliferative lesions could bring about the upper two-thirds of the epithelial tissue. In CIN3, the whole epithelium is proliferative. Treatment is usually given to remove abnormal cells in CIN2 and above [9,10]. The common symptom of cervical cancer includes abnormal vaginal discharge (which might be blood stained and foul smelling) and post-coital, postmenopausal, inter-menstrual, and excessive vaginal bleedings. Post-coital bleeding is regarded as a cardinal symptom [11]. Most often, the symptoms start when cervical precancerous lesions become malignant and begin to invade the surrounding tissues. This condition has a negative effect on the early treatment of cases in an area where there is limited awareness about cervical cancer among women. An effective approach capable of identifying precancerous lesions would be a good option to prevent morbidity and mortality [12]. A definitive diagnosis of CIN is by histopathological examination of cervical biopsy, and this most often prescribed following abnormal screening tests [13].

Persistence of HPV infection and further progression to cervical cancer is influenced by many factors. Early sexual intercourse, having multiple sexual partners, infection with a broad range of HPV types are important risk factors for HPV persistence [8]. Besides, tobacco smoking, high parity, long-term hormonal contraceptive use, co-infection with HIV, *Chlamydia trachomatis* and HSV-2, immunosuppression, and certain dietary deficiencies (like, folate), absence of regular screening, previous history of CIN are some of other identified factors elsewhere [14–18].

In most African countries, the fight against cervical cancer is still at its infancy because of poor vaccination coverage and lack of a well-functioning structure for the detection and management of precancerous lesions. There is very limited data on the histological profile of women in Ethiopia. Specifically, there is no study conducted on the histopathologic profile of cervical biopsy samples in the present study area. Therefore, this study aimed to determine the histopathological profile of women and risk factors associated with high-grade cervical lesions and cancer (CIN2+C) in northwest Ethiopia. The finding of the study could help to estimate the burden of different cervical lesion patterns among women visiting similar gynecology clinics in Ethiopia.

## Methods and materials

### Study setting

A hospital based cross-sectional study was conducted at Felege Hiwot Compressive Specialized Hospital (FHCSH) between 1 March 2019 and 30 Oct 2021. The hospital is located in Bahir Dar city, northwest Ethiopia, which is the capital of Amhara National Regional State, positioned about 565km away from Addis Ababa. The FHCSH, with more than 500 beds, is a tertiary health care facility that provides several types of specialized referral services for the people of Bahir Dar and other areas in northwest Ethiopia.

### Study population

In our study, 335 women presenting with suggestive signs and symptoms of abnormal cervical lesions including abnormal vaginal discharge and vaginal bleeding and those complaining of painful sexual intercourse were included [19,20]. The sample size was estimated by simple

proportion formula using the proportion of high grade squamous intraepithelial lesion at 27% among women attending the out-patient clinic of the department of Gynecology, Tikur-Anbessa Specialized Hospital, Ethiopia [21]. Participants with the following characteristics were considered to take part in the study: Age ≥30, had sexual history, were not pregnant, have an intact uterus and cervix, and not on menses. On the other hand, women who were seriously ill and those who were under treatment for invasive cervical cancer were excluded.

## Cervical sample collection

Interviewer-administered structured questionnaire was used to collect data on the participants' demographic, reproductive and gynecologic history. The tool was prepared following previous similar works [5,14–18,22]. Trained nurses who were working in the hospital gynecology department collected the questionnaire-based data. Each participant underwent general and pelvic examination in a compassionate and respectful process with informed consent. And then from the grossly visible lesion, a cervical swab was collected by a single-use broom-type brush (Digene HC2 DNA collection device: Qiagen, Hilden, Germany) whereas punch biopsies were collected using biopsy forceps by a licensed gynecologist following the recommended approach [23].

## Sample processing

Cervical swabs were used for the detection of high-risk Human papillomaviruses (HR-HPV) using the Abbott Alinity m system (Abbott Molecular, Des Plaines, IL, USA) following the manufacturer protocol at the Institute of Virology, Leipzig University Hospital, Germany.

With regard to the histological examination, once collected, cervical biopsies were placed in screw-capped and labeled bottles that contained 20ml of 10% formol-saline fixative solution and transported to the hospital pathology laboratory for downstream processing by the senior pathologist as described previously [24]. Lastly, the microscopic examination of slides was reported as normal histology, cervicitis, CIN1, CIN2, CIN3, cancer, and other miscellaneous findings. For this particular study, we used the acronym CIN2+C to represent high-grade precancerous lesions and cervical cancer together.

## Data analysis

Collected data entered and analyzed using SPSS version 25. Descriptive statistics was used to present the demographic, clinical characteristics and histopathological results of the study participants. A stepwise multivariable logistic regression model was used to find out factors associated with CIN2+C. Those variables with $p$-value $\leq 0.25$ in the bivariable logistic regression analysis were selected and entered into multivariable analysis [25]. A forward stepwise regression analysis was carried out. The model fitness was checked by the Hosmer-Lemeshow goodness of fit test value ($>0.5$). In multivariable analysis statistical significance was set at $p$-value less 0.05 to declare the risk factors considered in our study to be major explanatory factors associated with CIN2+C among studied women.

## Ethics approval and consent to participate

The study protocol was first approved by the institutional review board (IRB) of College of Health Sciences, Addis Ababa University (Protocol number: 087/19/DMIP) and then by the Ethiopian National Research Ethics Review Committee at the Ministry of Education (Ref number: 7/2-149/m259/35). Written informed consent was ensured from all study participants to take part in the study voluntarily after they get informed about the objective and purpose of the study.

## Results

### Clinical and histopathological features of the study participants

The detailed demographic and clinical related data of the study participants is presented in Table 1. In this study, 335 women who were attending the gynecology department at FHCSH were included. At enrolment, the study participants were in the age range of 30–80 years with a mean age of 46.5±11.4 years. The majority of the participants were married 262 (78.2%), housewives 298 (89.0%), were living permanently in rural settings 221 (66%), and did not have formal education 259 (77.3%). The mean body-mass index (BMI) of the participants was 23.3 ±3.5. About 31% of women were over weighted (BMI ≥ 25.0) and 5.2% were under weighted (BMI < 18.5).

The mean age at first sexual intercourse was at 15.7±2.6. The majority of the participants 265 (79.8%) were sexually active before the age of 18 years. Furthermore, 97.6%, 20.9%, 13.1%, 30.7%, and 50.9% of the participants had given birth at least once, were using hormonal contraceptive, were HIV positive, had a history of vaginal discharge and had multiple sexual partners, respectively. Among those participants who gave birth, 81.8% had more than two children, the mean number of children being five.

More than half of the participants 193(57.6%) were unaware of cervical cancer. Among those participants who had heard about cervical cancer, only 10(8.1%), knew that the disease is sexually transmitted. Likewise, the participants previous history of cervical screening practice and specifically screening in the last five years accounted for very low 93(27.8%) and 84 (25.1%), respectively. Among those participants who had previous screening practice, visual inspection with acetic acid (VIA) was the most common method by which they were screened 75 (91.5%), followed by a Pap test, 6 (7.3%). The prevalence of high-risk HPV (HR-HPV) was 54.3% (178/335) (Table 1).

With regard to the histological profile of the participants, the majority of cervical biopsy findings were found to be cervical cancer 140(41.8%; 95%CI: 36.6–47.1%). The proportion of high-grade precancerous lesions and cervical cancer together (CIN2+C), was 49.0% (164/335) (95%CI: 43.6–54.2%). Seventy-four (22.1%; 95%CI: 18–26.8) of the participants had normal histology. The detailed histological profile of the study participants is presented in Fig 1.

### Risk factors associated with high-grade cervical lesions among women seeking cervical cancer care

In bivariable logistic regression analysis, the participants' age group, number of deliveries, years of contraceptive use, having no information about cervical cancer, lifetime and the last five years of screening history and HR-HPV test results were found to be statically associated with CIN2+C ($p$-value <0.05) (Table 2). However, in our forward model of multivariable logistic regression analysis, only three variables were found to be statistically significant (p-<0.05), with respect to CIN2+C. These include being above the age of 50 years (AOR: 3.73; 95%CI: 1.80–7.82; $p$<0.001), having no screening history in the last five years (AOR: 2.03; 95% CI: 1.05–3.92; $p$ = 0.035) and being infected with HR-HPVs (AOR: 14.23; 95%CI: 7.9–25.64; p<0.001). A marginal statistical association was also observed for CIN2+C among those who had multiple sexual partners ($p$-value 0.07) (Table 3).

## Discussion

In the last few decades, a decreasing trend in the burden of cervical cancer was documented in the developed world. However, the disease remains a major public health problem in resource-limited settings, like Ethiopia [26,27]. The growing trends of identified risk factors, most often

**Table 1. Distribution of the study participants by their demographic and clinical characteristics.**

| Characteristics | | n (%) |
|---|---|---|
| Age groups (in years) | Mean, ±SD | 46.5, 11.4 |
| | 30–40 | 132 (39.4) |
| | 41–50 | 111 (33.1) |
| | >50 | 92 (27.5) |
| Permanent residence | Urban | 134 (34) |
| | Rural | 221 (66) |
| Marital status | Single | 4 (1.2) |
| | Married | 262 (78.2) |
| | Divorced | 33 (9.9) |
| | Other | 36 (10.7) |
| Educational status | No formal education | 259 (77.3) |
| | Primary | 42 (12.5) |
| | Secondary | 15 (4.5) |
| | Tertiary | 19 (5.7) |
| Type of occupation | House-wife | 298 (89.0) |
| | Private employee | 15 (4.5) |
| | Government employee | 20 (6) |
| | Other (prostitute) | 2 (0.6) |
| Body-mass index (BMI) | Mean, SD | 23.3, 3.5 |
| | Underweight (<18.5) | 16 (5.2) |
| | Normal (18.5–24.9) | 195 (63.9) |
| | Overweight (> = 25) | 94 (30.8) |
| Age at first sexual intercourse | Mean, SD | 15.7, 2.6 |
| | <18 years | 265 (79.8) |
| | ≥18 years | 67 (20.2) |
| Life-time number of sexual partners | Mean | 1.7 |
| | 1 | 163 (49.1) |
| | ≥2 | 169 (50.9) |
| Gave birth | Yes | 327 (97.6) |
| | No | 8 (2.4) |
| Number of deliveries | Mean, SD | 5.0, 2.4 |
| | 1–2 | 59 (18.2) |
| | 3–4 | 86 (26.5) |
| | ≥5 | 179 (55.3) |
| Use of hormonal contraceptive | Yes | 70 (20.9) |
| | No | 265 (79.1) |
| Years of hormonal contraceptive use | Mean, SD | 4.4, 3.0 |
| | 1–3 | 32 (56.1) |
| | 4–6 | 15 (26.3) |
| | >6 | 10 (17.5) |
| HIV sero-status | Positive | 44 (13.1) |
| | Negative | 286 (85.4) |
| | Unknown | 5 (1.5) |
| Previously treated for vaginal discharge | Yes | 103 (30.7) |
| | No | 232 (69.3) |
| Heard about cervical cancer | Yes | 142 (42.4) |
| | No | 193 (57.6) |

(*Continued*)

**Table 1.** (Continued)

| Characteristics | | n (%) |
|---|---|---|
| How cervical cancer transmitted? | Sexually | 10 (8.1) |
| | Do not know | 114 (91.9) |
| Mother or sister with cervical cancer history | Yes | 11 (3.3) |
| | No | 324 (96.7) |
| Tobacco smoking | Yes | 2 (0.6) |
| | No | 333 (99.4) |
| Living with a smoking person in same house | Yes | 3 (0.9) |
| | No | 232 (99.1) |
| Any kind of cancer history before | Yes | 1 (0.3) |
| | No | 334 (99.7) |
| History of cervical screening | Yes | 93 (27.8) |
| | No | 242 (72.2) |
| History of cervical screening in the last five years | Yes | 84 (25.1) |
| | No | 251 (74.9) |
| Used screening method | VIA | 75 (91.5) |
| | Pap | 6 (7.3) |
| | HPV DNA | 1 (1.2) |
| HR-HPV | Detected | 178 (54.3) |
| | Not Detected | 150 (45.7) |

SD: Standard deviation; VIA: Visual inspection with acetic acid; Pap test: Papanicolaou test; HR-HPV: High-risk Human papillomavirus.

diagnosis at its late stage, and poor knowledge about screening and other prevention approaches have amplified the burden of the cervical cancer in developing countries where there is limited vaccination and screening coverage [1,28]. Studies showed that the magnitude of precancerous lesions and cervical cancer were steadily increasing in Ethiopia [24,29]. Having data on the histological profile of cervical biopsies from institution-based studies, like ours, is important in revealing the overall picture of cervical lesions including cancer of the cervix to help guide concrete preventive works. The histopathological profile of cervical biopsy provides basis for a tailored management of patients and to establish a dependable prognosis [30].

To the best of our knowledge, this study is the first of its kind carried out to determine the histopathological profile of cervical biopsies and to explore important factors associated with CIN2+C among symptomatic women in northwest Ethiopia. Majority of cervical biopsies were diagnosed as cervical carcinoma, 41.8% and the proportion of CIN2+C was 49.0%. CIN-1 has high rates of spontaneous regression (>85%) within a couple of years, and it occasionally advances to cancerous lesions. However, if it is left untreated, CIN2 and CIN3 in particular could progress to cervical cancer [31]. The good thing is that, as the disease progresses slowly it gives enough time for early detection and management. The higher rate of premalignant lesions and cervical cancer in our finding implies that there is a need to increase awareness of the community and strengthen early cervical lesion detection and treatment.

Few similar studies conducted in Ethiopia reported comparable findings with our result. For instance, a retrospective study by Ameya et al. in Hawasa, reported the proportion of cancerous lesions to be 49.3% among symptomatic women in which precancerous and non-cancerous lesions were 9.9% and 32.4%, respectively [24]. Another histopathological based study in Addis Ababa, Ethiopia by Ergete et al. on the objective of assessing the causes of

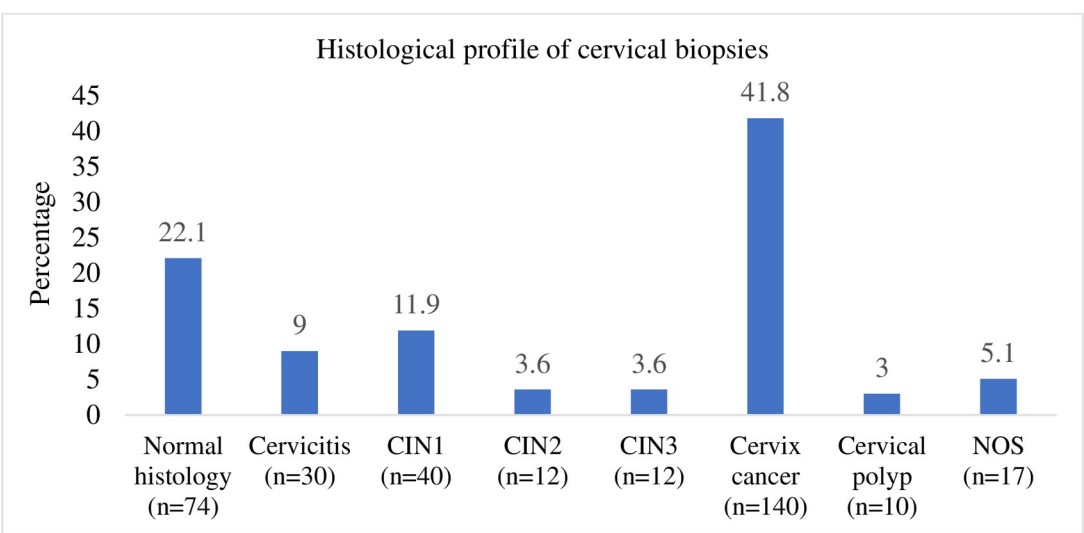

**CIN:** Cervical intraepithelial neoplasia.

**NOS:** Not otherwise classified; these categories include atypical endocervical glandular proliferation (1), focal of atypical columnar epithelium lining (1), unremarkable endocervical tissues (6), myoma (3), benign polypoid endocervical tissues (1), cervical wart (1), vulvar ulcer (1), napotian cyst (1), bloody and non-diagnostic sample (2).

**Fig 1. Percentage distribution of cervical biopsy histology findings.**

postmenopausal bleeding, reported that the proportion of cervical carcinoma was 84.8% amongst all malignant lesions of the genital tract [32]. The relatively higher proportion of cervical cancer in Ergete et al. study might be explained by the difference in the type of studied population in which they typically involved women with postmenopausal bleeding in contrast to our study which involved women aged ≥30 years.

Likewise, a nine years retrospective data analysis of 1,049 cervical histology reports in Ghana revealed that the majority of cervical samples at 99.4% were diagnosed as cervical carcinoma [27]. Another retrospective-based study aimed at analyzing 500 histological reports in Malawi showed that the proportion of cervicitis, endocervical polyp, and CIN to be 46.0%, 20.5%, and 24.4%, respectively [33]. Further, a study in Nepal on the histological profile of cervical samples indicated that benign, inflammatory, borderline, and malignant lesions were the commonest findings. Proportions of CIN1, CIN2, and CIN3 were 5.8%, 2.0%, and 0.4%, respectively among women who underwent cervical histological examination [11]. Similarly, in India, among 200 women with cervical histology examination, 35% had cervicitis, (12.5%) cervical polyps, (20%) CIN1, (3%) CIN2, (0.5%) CIN3, and 3.5% had cervical cancer [34]. The differences in the proportion of the histopathological findings across these studies in different parts of the world might be explained by different factors. These include variations in the severity of the lesions that women presented with during the time of examination, level of awareness of the participants for cervical screening practice, women health policy of nations, time and setting difference among the studies and socio-demographic related aspects.

At large, the relative higher proportion of high-grade and cancerous lesions in our report as compared to other countries could be due to the difference in type of population, difference in

**Table 2. Bivariable logistic regression analysis of factors associated with high-grade and cancerous lesions (CIN2+C).**

| Variables | | CIN2+C | | COR (95%CI) |
|---|---|---|---|---|
| | | Yes, n (%) | No, n (%) | |
| Age groups (in years) | 30–40 | 44(13.1) | 88(26.3) | 1 |
| | 41–50 | 53(15.8) | 58(17.3) | 1.82 (1.09–3.07) |
| | >50 | 67(20) | 25(7.5) | 5.36 (2.99–9.62) |
| Educational status | No formal education | 138(41.2) | 121(36.1) | 1.57 (0.61–4.03) |
| | Primary | 13(3.9) | 29(8.7) | 0.62 (0.20–1.89) |
| | Secondary | 5(1.5) | 10(3.0) | 0.69 (0.17–2.81) |
| | Tertiary | 8(2.4) | 11(3.3) | 1 |
| Type of occupation | House-wife | 151(45.1) | 149(44.5) | 1.88 (0.73–4.85) |
| | Private business | 6(1.8) | 9(2.7) | 1.24 (0.31–4.93) |
| | Government employee | 7(2.1) | 13(3.9) | 1 |
| Body-mass index (BMI) | Underweight | 11(3.6) | 5(1.6) | 1 |
| | Normal | 95(31.1) | 100(32.8) | 0.43(0.15–1.29) |
| | Overweight | 47(15.4) | 47(15.7) | 0.46(0.15–1.41) |
| Age at first sexual intercourse | <18 years | 137(41.3) | 128(38.6) | 0.59(0.34–1.02) |
| | ≥18 years | 26(7.8) | 41(12.3) | 1 |
| Life-time number of sexual partners | 1 | 74(22.5) | 89(27.1) | 1 |
| | ≥2 | 89(27.1) | 77(23.4) | 1.39(0.90–2.15) |
| Gave birth | Yes | 159(47.5) | 168(50.1) | 0.57 (0.13–2.41) |
| | No | 5(1.5) | 3(0.9) | 1 |
| Number of children | 1–2 | 20(6.2) | 39(12.0) | 1 |
| | 3–4 | 37(11.4) | 49(15.1) | 1.47(0.74–2.92) |
| | ≥5 | 101(31.2) | 78(24.1) | 2.53(1.34–4.67) |
| Use of hormonal contraceptive 0.03 | Yes | 26(7.7) | 44(13.1) | 0.54 (0.32–0.94) |
| | No | 138(41.2) | 127(37.9) | 1 |
| Years of hormonal contraceptive use | 1–3 | 9(15.8) | 23(40.4) | 1 |
| | 4–6 | 9(15.8) | 6(10.5) | 3.83 (1.06–13.91) |
| | >6 | 5(8.8) | 5(8.8) | 2.56 (0.59–11.00) |
| HIV sero-status | Positive | 22(6.6) | 22(6.6) | 1.05 (0.56–1.98) |
| | Negative | 142(42.4) | 149(42.4) | 1 |
| Treated for vaginal discharge | Yes | 47(14.0) | 56(16.7) | 0.83 (0.52–1.31) |
| | No | 115(34.3) | 117(34.9) | 1 |
| Heard about cervical cancer | Yes | 59(17.6) | 83(24.8) | 1 |
| | No | 105(31.3) | 88(26.3) | 1.68(1.08–2.60) |
| History of cervical screening | Yes | 32(9.6) | 61(18.2) | 1 |
| | No | 132(39.4) | 110(32.8) | 2.29(1.39–3.76) |
| History of cervical screening in the last five years | Yes | 27(8.1) | 57(17.0) | 1 |
| | No | 137(40.9) | 114(34.0) | 2.54(1.51–4.27) |
| HR-HPV detected | Yes | 134(40.9) | 44(13.4) | 14.52(8.44–25.00) |
| | No | 26(7.9) | 124(37.8) | 1 |

accessibility of better women health care practice including inadequate vaccination and cervical screening coverage and inadequate facilities for the management of precancerous lesions in Ethiopia. On top of this, the relatively higher prevalence of HIV in our setting might also be an important contributor to the high rate of cervical cancer in the present study. The prevalence of HIV in our study was 13.1%. Furthermore, as majority of our study participants lived in rural settings (66%) and had no formal education (77.3%), these might have a link with poor

**Table 3. Factors associated with high-grade and cancerous lesions (CIN2+C).**

| Variables | | CIN2+C | | COR (95%CI) | AOR (95%CI), *p*-value |
|---|---|---|---|---|---|
| | | Yes, n (%) | No, n (%) | | |
| Age groups (in years) | 30–40 | 44(13.1) | 88(26.3) | 1 | 1 |
| | 41–50 | 53(15.8) | 58(17.3) | 1.82 (1.09–3.07) | 1.40 (0.73–2.69), 0.316 |
| | >50 | 67(20) | 25(7.5) | 5.36 (2.99–9.62) | 3.73 (1.80–7.82), <0.001 |
| Lifetime number of sexual partners | 1 | 74(22.5) | 89(27.1) | 1 | 1 |
| | ≥2 | 89(27.1) | 77(23.4) | 1.39(0.90–2.15) | 1.69 (0.94–3.02), 0.07 |
| History of cervical screening in the last five years | Yes | 27(8.1) | 57(17.0) | 1 | 1 |
| | No | 137(40.9) | 114(34.0) | 2.54(1.51–4.27) | 2.03 (1.05–3.92), 0.035 |
| HR-HPV detected | Yes | 134(40.9) | 44(13.4) | 14.52(8.44–25.00) | 14.23(7.9–25.64), <0.001 |
| | No | 26(7.9) | 124(37.8) | 1 | 1 |

HR-HPV; High-risk Human papillomaviruses; CI: Confidence interval; COR: Crude odds ratio; AOR: Adjusted odds ratio.

The Hosmer-Lemeshow model fitness test result; Chi-square; 3.47, df; 8, significance; 0.90.

awareness and knowledge among studied women about cervical cancer including its means of transmission, prevention and the stage when they should seek medical attention. More than half of the participants in our study also reported that they were unaware of cervical cancer at all. Previous studies also stated that most Ethiopian women had poor knowledge about cervical cancer [35,36].

The proportion of cervical screening was quite low (27.8%) in our study. A meta-analysis by Desta et al. revealed that cervical cancer screening among eligible women in Ethiopia was below 15%. The lowest national screening coverage was documented in Amhara Regional State where the present study was carried out [37]. A study in Addis Ababa, Ethiopia reported that absence of signs and symptoms in the early period of cervical precancerous lesions and inadequate awareness about screening were the main perceived reasons among women who do not have previous screening practice [38]. These all factors could possibly explain the reasons behind the higher proportion of high-grade and cancerous lesions (CIN2+C) among women in the present study area.

Studies in different parts of the world identified important predictors for CIN2+C although there was inconsistency among reports. Some of the commonly identified factors include younger age at first sex and pregnancy, having multiple sexual partners, increasing period of contraceptive use, cigarette smoking, high parity, low socioeconomic status, and immunosuppression. Indeed, these factors either increased the risk of exposure to HR-HPV infection or might predispose for a compromised immune response to the infection [39–46].

In our study, the multivariable logistic regression analysis showed that women >50 years of age were about four times more likely to develop CIN2+C than the young groups. Similarly, those who were not screened and those with HR-HPVs infection were about two and fourteen times more likely to develop CIN2+C, respectively. However, most of the variables tested in the model did not show an independent statistical association with CIN2+C. This might be because of a sample size matter, the role of chance and confounding variables influencing each other.

The proportion of cervical cancer steadily increased with the age of study participants. From 21.2% at the age of 30–40 years, the proportion of cervical cancer increased to almost 71.3% among postmenopausal women above the age of 50 years (*p*-value <0.0001). Ameya and his colleague from Hawassa, Ethiopia also reported exactly the same pattern matching our report [24]. A study in Jimma, Ethiopia also revealed that women above 50 years were more

likely to have advanced cervical lesions [47]. A study in Ghana also revealed that 71% of the patients with cervical cancer were ≥50 years old [27], which is precisely in line with our report. Similarly, a study in Nepal reported that cervical cancer was primarily recorded among women aged 51–60 years [11].

Studies in Ethiopia showed that, patients with cervical cancer were most often visited health facilities after a long time leading to advanced stages of the disease at time of diagnosis [47,48]. This might be due to poor awareness of the disease and the nature of the screening program in the country that is mainly by using VIA technique. In developing countries like Ethiopia, where there is no well-established cervical screening and vaccination coverage, such kind of disease pattern across age categories is highly predictable. The best way to avert the continued increase of the disease with age would be through comprehensive cervical screening of eligible women with convenient method. A trial by Gizaw et al. demonstrated a significantly higher levels of population-based cervical screening uptake and adherence for self-collection HPV testing than the VIA based screening in Ethiopia [49].

In the present study, the other important predictor for CIN2+C was infection with HR-HPV. We noted that the proportion of HR-HPV increased with degree of the lesion (*p*-value <0.0001). Studies in South Africa and Brazil also reported findings in line with our report [46,50]. In our study, 265 (79.8%) of the participants started sex before the age of 18 years. The mean age at first sexual intercourse was 15.7± 2.6 years. Moreover, 30.7% and 50.9% of the participants had history of vaginal discharge and sex with multiple partners. These all findings indirectly suggest the participants' risk of exposure for different kinds of sexual transmitted infections including HR-HPVs that proportionally increase the risk of high-grade lesions and eventually to cancer development. It is a well establish knowledge that women who have persistent infection with HR-HPVs ultimately develop CINs including cancer in later years [51].

Lastly, we noted that women who had no cervical cancer screening practice in the last five years were about two times more likely to develop CIN2+C. Some women not screened regularly could potentially develop cancerous lesions at the late stage of their life. In contrast, a study showed that sufficiently screened women were less likely to be diagnosed with high-grade and cancerous lesions [52]. Since the majority of our study participants were from rural settings, had no formal education, and without information about cervical cancer, they might miss important cervical cancer prevention approaches, including screening at regular interval. In fact, illiteracy among women results in poor health-seeking behavior [53], especially for gynecologic-related symptoms, which most people in Ethiopia considered it as a taboo. In the present study, participants' previous history of cervical screening practice was reportedly very low, at 27.8%. Screening encompasses testing for HPV infection to detect precancerous lesions the earliest possible, followed by treatment when appropriate. When screening detects HPV infection or precancerous lesions, these can easily be treated and therefore advanced stages of the lesion could be prevented. The WHO recommends that screening should start from 30 years of age in the general population, with a regular screening every 5–10 years, and from 25 years of age for HIV-positive women who should be screened frequently, every 3–5 years. Vaccination, screening, and treatment of precancerous lesions are the cost-effective ways to prevent cervical cancer [54].

As far as associated factors for CIN2+C are concerned, the clinical implication of our findings implies that women above the age of 50, those who tested positive for HR-HPVs, and those who had no cervical screening history should get top priority for close follow-up.

In this study, we provided the first histopathological-based cervical biopsy report among the study participants in northwest Ethiopia. However, our findings should be interpreted in line with the following limitations. We have used punch biopsy taken blindly in four quadrants

i.e. it is not colposcopy guided hence there is a chance for inadequate sampling. The histopathological report was generated by a single pathologist i.e. slides were not double-checked by another pathologist.

## Conclusions

Majority of the study participants lived in the rural setting, had no formal education and unaware of cervical cancer. The level of CIN and cervical cancer was high in this study. Age above 50 years, HR-HPV positive cases and no screening practice in the last five years were found significantly associated with high-grade and cancerous lesions. This calls for urgent action to expand and strengthen the ongoing strategies in increasing the awareness of the public about cervical cancer in Ethiopia in general and in Amhara Regional State in particular. Women who are at odds of increased risk for cervical cancer should be regularly tested including checking for HR-HPV infection at the time of screening.

## Acknowledgments

First, we would like to acknowledge the staff at the gynecology department of FHCSH for their kind cooperation during the data collection process. Second, our special gratitude goes to Dr. Eva J. Kantelhardt at the Department of Gynecology, Martin-Luther-University, Halle-Wittenberg, Germany for her generous support. Finally, the authors would like to thank the study participants and the data collectors.

## Author Contributions

**Conceptualization:** Awoke Derbie, Tamrat Abebe.

**Data curation:** Bereket Amare, Eyaya Misgan, Melanie Maier.

**Methodology:** Yimtubezinash Woldeamanuel.

**Writing – original draft:** Awoke Derbie.

**Writing – review & editing:** Eyaya Misgan, Endalkachew Nibret, Melanie Maier, Yimtubezinash Woldeamanuel, Tamrat Abebe.

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
