## [Decision Letter · Decision Letter 0]

4 Aug 2022

PONE-D-22-10841Histopathological profile of cervical punch biopsies and risk factors associated with high-grade cervical precancerous lesions and cancer in northwest EthiopiaPLOS ONE

Dear Dr. Derbie,

Thank you for submitting your manuscript to PLOS ONE. After careful consideration, we feel that it has merit but does not fully meet PLOS ONE’s publication criteria as it currently stands. Therefore, we invite you to submit a revised version of the manuscript that addresses the points raised during the review process.

The manuscript has been evaluated by two reviewers, and their comments are available below.

The reviewer has raised a number of concerns related to reporting of the study methodology and results. In particular, the reviewers had requested additional information regarding samples size calculations, participant recruitment as well as the current cervical screening practices in the study location.

It may be seen that one of the reviewers had commented on the scientific impact of the study. While PLOS ONE does not issue decisions based on perceived novelty we do however require studies to further contribute to scientific knowledge, https://journals.plos.org/plosone/s/criteria-for-publication#loc-2. As such we suggest discussing related literature and how the current study further contributes to scientific knowledge.

Finally, one of the publication criteria at PLOS ONE (https://journals.plos.org/plosone/s/criteria-for-publication#loc-5 )is that articles must be presented in an intelligible fashion and written in clear, correct, and unambiguous English. Both reviewers had raised concerns regarding the quality of English language of the mansucript and we recommend thoroughly copy editing the mansucript for language presentation.  

We look forward to receiving your revised manuscript.

Kind regards,

Lucinda Shen, MSc

Staff Editor

PLOS ONE

Journal Requirements:

4.Thank you for stating the following financial disclosure:

Reviewers' comments:

Reviewer's Responses to Questions

**Comments to the Author**

1. Is the manuscript technically sound, and do the data support the conclusions?

Reviewer #1: Partly

Reviewer #2: Yes

2. Has the statistical analysis been performed appropriately and rigorously? 

Reviewer #1: No

Reviewer #2: Yes

3. Have the authors made all data underlying the findings in their manuscript fully available?

Reviewer #1: Yes

Reviewer #2: Yes

4. Is the manuscript presented in an intelligible fashion and written in standard English?

Reviewer #1: Yes

Reviewer #2: Yes

5. Review Comments to the Author

Reviewer #1: Comments to the authors

While it would be of some interest, this study does not bring any breakthrough to the scientific community. Generally, the manuscript has a lot of grammatical errors and inappropriate use of vocabulary.

The specific comments include:

Background

1. The data reported in P3L66-67 should be cited.

2. Sentences expressing feelings need to be avoided in your writings. For instance, “….. interesting report….” P3L76.

3. The second paragraph is missing the state-of-the-art description of the burden of cervical cancer and associated prevention program in Ethiopia, especially the screening program. In addition, appropriate references related to Ethiopia are missing (P3L74-79). This should be rewritten.

4. P6L145-154 should be summarized and shifted to the background section.

5. The purpose of the study is not clear. I guess the authors wanted to identify the risk factors associated with the burden of cervical cancer in the studied population. The question is what did they achieve with this study? The conclusion being vague is not giving any feasible answer.

Materials and Methods

6. The authors need to revise the histopathological classification of cervical cancer cases according to the World Health Organization classification of tumors from women reproductive organs. The subtypes are benign, precancerous, and cancerous cervical lesion. The following reference might be of help in revising your work: -

Zuberi, Z., Mremi, A., Chilongola, JO., Semango, G., and Sauli, E. (2021). Expression of p16 and TOP2A biomarkers associated with clinico-histopathological features among women seeking cervical cancer care in Tanzania. PLoS One, 16 (10): e0259096

Results

7. The authors should report the mean age as mean ± SD and not mean, SD in Table 1.

8. The subheadings of demographic characteristics, the histological profile of the participants, and reproductive and gynecological features can be merged into a new subheading titled “Clinical histopathological features among women seeking cervical cancer care.” In addition, the subheading P10L209 should be renamed to “Risk factors associated with cervical lesions among women seeking cervical cancer care.”

9. The incorrect in-text citation of the table at P8L195 should be Table 2, not Table 1.

10. The findings presented at L181-219 should be supported by the biological relevance of the study observations.

11. Why statistical significance was not presented in Table 2? That’s to say, include the p-values.

12. In Table 2, the univariate logistic regression could be performed for benign, precancerous, and cancerous cervical lesions to identify the risk factors associated with cervical cancer lesions.

13. What exclusion and inclusion criteria were deployed to combine high-grade (CIN-2) and cervical cancer lesions? CIN-3 is also a high-grade cervical lesion, why was excluded?

14. Why a p-value ≤ 0.25 was considered for univariate regression analysis (P7L159-160)?

15. Figure 1 should be improved by following the histopathological classification of cervical cancer cases according to the World Health Organization classification of tumors from women reproductive organs by Kurman et al. 2014.

Discussion

16. The discussion should be completely rewritten. In addition, the discussion is repeating the results that is unnecessary.

Reviewer #2: The paper "Histopathological profile of cervical punch biopsies and risk factors associated with high-grade cervical precancerous lesions and cancer in northwest Ethiopia" aimed to determine the histopathological profile of 335 women and risk factors associated with high-grade cervical lesions and cancer over a period of 2.5 years in northwest Ethiopia. It is necessary to do this type of work, to publicize the cervical characteristics of the patients as well as unifying criteria between pathologists.

The paper is interesting and fairly well written. The data which are presented are moderate to clearly presented though there are several issues that deserve more attention or should be improved.

Specific points:

Methods section

1. The authors should give some more information on how current screening in Ethiopia or area of study is organized. Are cervical scrapes performed? Is there any way of screening with pap-smears after which women are referred for colposcopy with cervical biopsy? This will give the reader a better understanding why cervical punch biopsies were taken blindly in the current study.

2. This being a prospective study, authors should explain how sample size was determined? The authors should clearly explain how they reached at a sample size of 335.

Results section

1. It is good to note that Table 1 has both demographic and clinical characteristics. What does Bahir Dar 2021 on line number 197 and all the tables titles mean? Unless it is in agreement with the journal requirement I recommend that the authors delete it from all the tables, because it does not make sense.

In Table 1

o you report 335 participants but the sum of life time number of sexual partners is 332 (163 + 169=332) less than total participants, number of deliveries is 342 (59+86+197=342) exceeding number of participants in the study.

o It is also indicated that there were 70 participants who reported to have used hormonal contraceptives but the breakdown for years of hormonal contraceptive use is 57 (32+15+10 =57).

o The women who had heard about cervical cancer are 142 but when they were asked how it is transmitted only 124 (10+114=124) responded. Some are missing

o The same applies to those screened for cervical cancer in the last 5 years and high-risk Human papillomavirus (HR-HPV) detection

Conclusion

1. Even though the majority of the study participants lived in the rural setting (66%), had no formal education (77.3%) and never heard about cervical cancer (57.6%), the conclusion does not specifically address this population.

Minor issues

1. Authors should proof read the manuscript because there are a few typos and language that need to be corrected. For example in results section line number 174, in discussion section line number 240, 245, 324 amongst others.

2. Authors also need to check that references, words, symbols and stats are not running into each other. For example words vs references are in background section line number 72, discussion section line number 235 etc

6. PLOS authors have the option to publish the peer review history of their article (what does this mean?). If published, this will include your full peer review and any attached files.

Reviewer #1: No

Reviewer #2: **Yes: **Paul Uchizi Kaseka

---

## [Author Response · Author response to Decision Letter 0]

20 Aug 2022

Comments to the Author

Reviewer #1: Comments to the authors

While it would be of some interest, this study does not bring any breakthrough to the scientific community. Generally, the manuscript has a lot of grammatical errors and inappropriate use of vocabulary.

• Dear reviewer, thank you so much for your important comments and suggestion. We have gone through the entire paper and we made all possible corrections in the revised version of the manuscript. 

The specific comments include:

Background

1. The data reported in P3L66-67 should be cited.

• Thank you, a correction has been made. 

2. Sentences expressing feelings need to be avoided in your writings. For instance, “….. interesting report….” P3L76.

• Thank you, correction has been made. 

3. The second paragraph is missing the state-of-the-art description of the burden of cervical cancer and associated prevention program in Ethiopia, especially the screening program. In addition, appropriate references related to Ethiopia are missing (P3L74-79). This should be rewritten.

• Thank you, corrections have been made at this point. Kindly check the second paragraph. 

4. P6L145-154 should be summarized and shifted to the background section.

• Thank you, the summary of this part is shifted to the background section. 

5. The purpose of the study is not clear. I guess the authors wanted to identify the risk factors associated with the burden of cervical cancer in the studied population. The question is what did they achieve with this study? The conclusion being vague is not giving any feasible answer.

• Dear reviewer, as we have clearly indicated in the last paragraph of the background section, the objective of this study is to determine the histopathological profile of women and risk factors associated with high-grade cervical lesions and cancer (CIN2+C) in northwest Ethiopia. It is not about burden of cervical cancer among the studied population.

Materials and Methods

6. The authors need to revise the histopathological classification of cervical cancer cases according to the World Health Organization classification of tumors from women reproductive organs. The subtypes are benign, precancerous, and cancerous cervical lesion. The following reference might be of help in revising your work: -

Zuberi, Z., Mremi, A., Chilongola, JO., Semango, G., and Sauli, E. (2021). Expression of p16 and TOP2A biomarkers associated with clinico-histopathological features among women seeking cervical cancer care in Tanzania. PLoS One, 16 (10): e0259096

• Dear reviewer, thank you for the reference. According to the provided citation in Table 1, the authors tried to classify the histopathological features into three main groups (benign, precancerous, and cancerous lesions) and they also further classified these into subgroups, which is in line with our classification scheme. To appreciate the overall histological profile of cervical samples from women, such an explicit approach is much better than presenting the histological report in only three groups. On top of this, there are histological findings that cannot be accommodated in these groups (like the normal histology and the NOS groups). Kindly note that there is no single best approach to classify cervical histological findings into different groups. However, if still the reviewer insists to present our cervical histology data only in three groups, we can do that in the later version of the manuscript. 

Results

7. The authors should report the mean age as mean ± SD and not mean, SD in Table 1.

• Thank you, we have revised this.

8. The subheadings of demographic characteristics, the histological profile of the participants, and reproductive and gynecological features can be merged into a new subheading titled “Clinical histopathological features among women seeking cervical cancer care.” In addition, the subheading P10L209 should be renamed to “Risk factors associated with cervical lesions among women seeking cervical cancer care.”

• Thank you, corrections have been made. Kindly check this in the revised version of the result section. 

9. The incorrect in-text citation of the table at P8L195 should be Table 2, not Table 1.

• Dear reviewer, kindly check that the citation is correct. The statement that states the prevalence of high-risk HPV (HR-HPV) at 54.3% (178/335) is presented in Table 1 (not in Table 2).

10. The findings presented at L181-219 should be supported by the biological relevance of the study observations.

• Dear reviewer, the findings reported in this part (L181-219 ) of the first submission represents the reproductive and gynecology related features and important risk factors that predict high-grade cervical lesions and cancer. The variables stated in this section were adopted from previous similar works and they are biologically relevant with our outcome variables. 

11. Why statistical significance was not presented in Table 2? That’s to say, include the p-values.

• Dear reviewer, in Table 2 we have indicated the crude bivariable logistic regression analysis where we learn about statstical significance indirectly from the provided confidence interval. However, we have presented the p–value in in our multi-variable logistic regressio analysis part (Table 3) apart from the confidence interval. 

12. In Table 2, the univariate logistic regression could be performed for benign, precancerous, and cancerous cervical lesions to identify the risk factors associated with cervical cancer lesions.

• Dear reviewer, one of the objective of our study is to find out factors associated with high-grade cervical lesions and cancer. We did multivariable analysis for CIN2, CIN3 and cancer (the so-called CIN2+C lesions). However, due to their limited clinical significance we were not interested to run multivariable logistic regression for benign lesions. 

13. What exclusion and inclusion criteria were deployed to combine high-grade (CIN-2) and cervical cancer lesions? CIN-3 is also a high-grade cervical lesion, why was excluded?

• Dear reviewer, thank you for the question. Kindly note that CIN2+C referes to CIN2, CIN3 and cervical cancer cases together. Hence, the analysis in Table 3 didn‘t exclude CIN3, rather it is part of the analysis. 

14. Why a p-value ≤ 0.25 was considered for univariate regression analysis (P7L159-160)?

• Dear reviewer, according to the reference we have cited at this point of the first manuscript submission, P7L160, (Bursac Z, Gauss CH, Williams DK, Hosmer DW. Purposeful selection of variables in logistic regression. Source Code Biol Med. 2008;3:17-. doi: 10.1186/1751-0473-3-17. ), p-value of 0.2-0.25 can be considered to select variables for multivariable logistic regression test based on the clinical or statistical significance. Hence, we took 0.25 to select variables in bivariable analysis for the multivariable logistic regression considering their biological/clinical significance. 

15. Figure 1 should be improved by following the histopathological classification of cervical cancer cases according to the World Health Organization classification of tumors from women reproductive organs by Kurman et al. 2014.

• Dear reviewer, our explanation for your question on Figure 1 is same as we presented for Q #6. Kindly check this above. 

Discussion

16. The discussion should be completely rewritten. In addition, the discussion is repeating the results that is unnecessary.

• Thank you, it would have been fine if you had indicated the specific areas of this section that needs rewritten. However, after reading the section between a line we made all necessary amendments to make it more readable. 

Reviewer #2: The paper "Histopathological profile of cervical punch biopsies and risk factors associated with high-grade cervical precancerous lesions and cancer in northwest Ethiopia" aimed to determine the histopathological profile of 335 women and risk factors associated with high-grade cervical lesions and cancer over a period of 2.5 years in northwest Ethiopia. It is necessary to do this type of work, to publicize the cervical characteristics of the patients as well as unifying criteria between pathologists.

The paper is interesting and fairly well written. The data which are presented are moderate to clearly presented though there are several issues that deserve more attention or should be improved.

• Thank you, your positive feedback about the paper is highly appreciated!

Specific points:

Methods section

1. The authors should give some more information on how current screening in Ethiopia or area of study is organized. Are cervical scrapes performed? Is there any way of screening with pap-smears after which women are referred for colposcopy with cervical biopsy? This will give the reader a better understanding why cervical punch biopsies were taken blindly in the current study.

• Dear reviewer, thank you for your comments and questions. Due to a number of reasons (especially, because of the absence of basic infrastructure for histo-pathological examination, including pathologists) currently cervical cancer screening in Ethiopian is performed using the VIA technique in almost all public hospitals and facilities. We have indicated this in the background and discussion section of the revised manuscript. 

• However, there are private settings that provide pap-smear and HPV-based tests. In our study, we recruited women for punch biopsy examination just to see the distribution of histological profiles among women presented with different cervical abnormalities and in connection to find out important risk factors that could predict high-grade cervical lesions and cancer to help guide public health intervention measures. 

2. This being a prospective study, authors should explain how sample size was determined? The authors should clearly explain how they reached at a sample size of 335.

• Dear reviewer, the sample size was estimated by using the proportion of high-grade cervical lesion at 27% among women attending the outpatient clinic of the department of Gynecology, Tikur-Anbessa Hospital (Ethiopia), at 5% margin of error and 95% confidence level. We have added 10% non-response rate. We have consulted this citation to take the proportion (Wolday, D., et al., HPV genotype distribution among women with normal and abnormal cervical cytology presenting in a tertiary gynecology referral Clinic in Ethiopia. Infect Agent Cancer, 2018. 13(28): p. 018-0201). A revision has been made in the manuscript. Kindly check this.

Results section

1. It is good to note that Table 1 has both demographic and clinical characteristics. What does Bahir Dar 2021 on line number 197 and all the tables titles mean? Unless it is in agreement with the journal requirement I recommend that the authors delete it from all the tables, because it does not make sense.

• Thank you, we deleted that. We put Bahir Dar and 2021 in each table to make the table self-explanatory in which the ‘what’, ‘where’ and ‘when’ can easily read from the tables. 

In Table 1

o you report 335 participants but the sum of life time number of sexual partners is 332 (163 + 169=332) less than total participants, number of deliveries is 342 (59+86+197=342) exceeding number of participants in the study.

o It is also indicated that there were 70 participants who reported to have used hormonal contraceptives but the breakdown for years of hormonal contraceptive use is 57 (32+15+10 =57).

o The women who had heard about cervical cancer are 142 but when they were asked how it is transmitted only 124 (10+114=124) responded. Some are missing

o The same applies to those screened for cervical cancer in the last 5 years and high-risk Human papillomavirus (HR-HPV) detection

• Dear reviewer, thank you so much for the insightful comment. For some of the variables, the discrepancy in figures happened because of missing data while the questioner based items were collected by nurses. 

• There was typing error while transcribing the number of deliveries into the table. Now, it is revised in Table 1.

Conclusion

1. Even though the majority of the study participants lived in the rural setting (66%), had no formal education (77.3%) and never heard about cervical cancer (57.6%), the conclusion does not specifically address this population.

• Dear reviewer, thank you. We have revised this section based on your important suggestion. Kindly check this in the conclusion part of the revised document.

Minor issues

1. Authors should proof read the manuscript because there are a few typos and language that need to be corrected. For example in results section line number 174, in discussion section line number 240, 245, 324 amongst others.

• Thank you, we made all the necessary corrections to make the revised paper readable at these lines and others. 

2. Authors also need to check that references, words, symbols and stats are not running into each other. For example words vs references are in background section line number 72, discussion section line number 235 etc

• Thank you, we have checked that. 

With kind regards!

---

## [Editor Report · Decision Letter 1]

30 Aug 2022

Histopathological profile of cervical punch biopsies and risk factors associated with high-grade cervical precancerous lesions and cancer in northwest Ethiopia

PONE-D-22-10841R1

Dear Dr. Awoke Derbie

We’re pleased to inform you that your manuscript has been judged scientifically suitable for publication and will be formally accepted for publication once it meets all outstanding technical requirements.

Kind regards,

Paul Uchizi Kaseka;, MSN

Guest Editor

PLOS ONE
---

## [Editor Report · Acceptance letter]

2 Sep 2022

PONE-D-22-10841R1 

Histopathological profile of cervical punch biopsies and risk factors associated with high-grade cervical precancerous lesions and cancer in northwest Ethiopia 

Dear Dr. Derbie:

I'm pleased to inform you that your manuscript has been deemed suitable for publication in PLOS ONE. Congratulations! Your manuscript is now with our production department. 

Kind regards, 

on behalf of

Dr. Paul Uchizi Kaseka; 

Guest Editor

PLOS ONE